# Characterization of Organosolv Lignins and Their Application in the Preparation of Aerogels

**DOI:** 10.3390/ma15082861

**Published:** 2022-04-13

**Authors:** Piia Jõul, Tran T. Ho, Urve Kallavus, Alar Konist, Kristiina Leiman, Olivia-Stella Salm, Maria Kulp, Mihkel Koel, Tiit Lukk

**Affiliations:** 1Department of Chemistry and Biotechnology, Tallinn University of Technology, Akadeemia tee 15, 12618 Tallinn, Estonia; piia.joul@gmail.com (P.J.); thihol@taltech.ee (T.T.H.); kristiina.leiman@taltech.ee (K.L.); olivia-stella.salm@taltech.ee (O.-S.S.); maria.kulp@taltech.ee (M.K.); mihkel.koel@taltech.ee (M.K.); 2Department of Mechanical and Industrial Engineering, Tallinn University of Technology, Ehitajate tee 5, 19086 Tallinn, Estonia; urve.kallavus@taltech.ee; 3Department of Energy Technology, Tallinn University of Technology, Ehitajate tee 5, 19086 Tallinn, Estonia; alar.konist@taltech.ee

**Keywords:** aerogel, lignin, hydrogels, functional polymer, biomass, supercritical drying

## Abstract

The production of novel materials and value-added chemicals from lignin has received considerable attention in recent years. Due to its abundant occurrence in nature, there is a growing interest in utilizing lignin as a feedstock for functional materials production, for example aerogels. Much like in the synthesis of phenol-based resins, the vacant ortho positions of the aromatic rings in lignin can crosslink with formaldehyde and form polymeric gels. After drying the hydrogels with supercritical CO_2_, highly porous aerogels are obtained. Current study focuses on the preparation and thorough parametrization of organosolv lignins from different types of lignocellulosic biomass (aspen, pine, and barley straw) as well as their utilization for the preparation of lignin-5-methylresorcinol-formaldehyde aerogels. The thorough structural characterization of the obtained aerogels was carried out by gas adsorption, IR spectroscopy, and scanning electron microscopy. The obtained lignin-based monolithic mesoporous aerogels had specific surface areas and total pore volumes in the upward ranges of 450 m^2^/g and 1.4 cm^3^/g, respectively.

## 1. Introduction

Lignocellulosic biomass is the most abundant renewable resource on Earth. This complex resource is composed of three major biopolymers-lignin, cellulose, and hemicellulose. Proportionally, lignin makes up 10–25% of its dry mass by weight, while the other counterparts–cellulose and hemicellulose, amount to approximately 40–50% and 20–30%, respectively. The topic has been thoroughly researched and many good review articles are available on this subject [1]. However, current biorefinery processes are focused primarily on the utilization of the carbohydrate fractions (cellulose and hemicellulose), while lignin remains underutilized [2]. Only a small percentage, less than 2%, of the approximately 70 million tons of lignin that are produced during pulping by the paper industry is utilized, primarily as concrete additives, stabilizing agents, or dispersants and surfactants. The remainder is simply discarded as waste or burnt as low-grade fuel. Similarly, only a fraction of the lignin portion that is considered a co-product of cellulosic ethanol production is used, and primarily for thermal energy production.

Lignin is an insoluble, highly branched, and randomly structured amorphous polymer. The typical lignin content in softwoods is 24–33%, 19–28% in hardwoods, and 15–25% in cereal straws, bamboo, and bagasse. Chemically speaking, lignin is a highly complex polyphenol, consisting of various methoxylated phenylpropane structures, which can be considered as the most abundant natural source of aromatics [3]. Although the exact polymeric structure of lignin is not well defined, the key substructures and linkages are known, and on this basis possible structures can be proposed. There are three primary phenylpropanoid monomer units in lignin: syringyl-(S), guaiacyl-(G), and *p*-hydroxyphenyl-(H) units that are derived from the respective monolignols–sinapyl, coniferyl, and *p*-coumaryl alcohols (Figure 1). The ratio between these units, the molecular weight of lignin, and the proportion of lignin to cellulose and hemicellulose differ from species to species significantly [4].

The effective extraction and separation of high purity lignin from lignocellulosic biomass is crucial for its efficient valorization. The rupture of bonds to separate lignin from carbohydrates and its partial depolymerization to make it extractable is required for its solubilization in proper solvents for further upgrading and processing. While the efficacy of lignin depolymerization is dependent on many different parameters (solvent, temperature, catalysts, etc.), the fragmentation of lignin is achieved primarily through the cleavage of β-O-4 and 4-O-5 aryl ether linkages.

Currently, the majority of lignin is produced by the pulp and paper-making industry–most commonly by Kraft, sulphite, or soda pulping, but also via hydrolysis/fractionation by hot water, dilute acid, or alkali, organic solvents or via enzymatic hydrolysis in a biorefinery process [2,5]. From these, soda, organosolv and hydrolysis lignins are sulfur-free materials [6].

Some technologies which are currently utilized primarily in research laboratories or in pilot plant reactors must be mentioned: (i) hydrolysis lignin that is obtained from the saccharification process of hemicellulose and cellulose during cellulosic ethanol production, and (ii) lignins from lignocellulose processing with molten salts (ionic liquids and deep eutectic solvents). Studies on using molten salts to develop more environmentally friendly processes for the fractionation of lignocellulosic biomass are gaining more and more attention [7].

In general, isolated lignins vary in structure due to the source of feedstock (i.e., type of biomass), extraction methods, and processing severity. Additionally, controlled degradation of lignin that leads to uniform starting material is necessary to arrive at a desired product in downstream valorization processes [8,9,10].

Biopolymer-based materials are suitable for the fabrication of biopolymer-based gels [11]. Phenolic resins are well-known polymeric materials where by manipulating the phenolic (or substituted phenols) to aldehyde monomer ratio and the amount of solvent, pH, catalyst type, reaction temperature, and reaction time, give rise to a variety of resin structures with a wide range of possible properties. In 1989, Pekala performed the aqueous polycondensation of resorcinol (1,3-dihydroxybenzene) with formaldehyde (FA) as a synthetic route to organic aerogels-low-density, open-porous, solid framework of a gel that was isolated intact from the gel’s liquid component and have pores in the range of <1 to 100 nm in diameter. In this system, resorcinol acted as the multifunctional monomer and FA formed methylene and/or methylene ether bridges between the benzene rings [12]. Some advantage in the preparation process was achieved with the replacement of resorcinol with 5-methylresorcinol (5-MR) [13,14]. The phenolic and polymeric nature of lignins make them potent replacements for phenol/resorcinol in a multitude of industrial applications as biopolymers with considerable economic and environmental benefits.

The presence of a free hydroxyl group and the vacant ortho- or para-sites on the aromatic ring are a prerequisite to polycondensation reaction, which is necessary for gel formation. In lignin, the H- and G-units possess the ortho reactive site that is similar to resorcinol, which can crosslink with FA. Properties such as these can be useful for the effective utilization of lignin as a phenol replacement in lignin-phenol-formaldehyde resins [15].

Conversely, phenolic groups in lignin are often substituted in the para position by an aliphatic chain. Here, the H-type unit possess more than one activatable site, offering the potential to create highly crosslinked structures. In G-type units, one ortho position is already occupied by a methoxy group, whereas in S-type units both ortho-positions are occupied. This higher degree of substitution results in a significantly lower reactivity of lignin as compared to phenol [16]. It is clear that the H and G-units are able to react with FA in the synthesis of lignin-based resins while the S- (no activatable site is present) unit is not [17]. In this sense, softwood lignins (containing mainly G-units) are a better choice for lignin-based resin synthesis than hardwood lignins (containing S-units and some H-units). Alkali and organosolv lignins with a higher number of G-units, a free ortho position, and moderate molecular weight seem to be better candidates for the synthesis of 100% lignin-based resins [16]. Therefore, in the synthesis of lignin-based aerogels, due to the nature of the reaction between lignin and FA, the sub-structures within lignin play a more important role than either molecular weight or polydispersity of lignin [18]. Despite that, lignins are less reactive towards addition/substitution reactions due to the lack of reactive sites [19]. The replacement of phenol with lignin would provide a cost-effective and green alternative as phenolic resins are used in many industrial applications (e.g., automotive, computing, aerospace, and construction), but also in the manufacturing of engineered wood products [16]. The control over lignin functionality appears to be one of the most important challenges for the development of lignin-based materials, and a multitude of analytical chemistry and spectroscopic tools (e.g., IR and NMR) are needed to parametrize the particular lignin sample structure.

Similar to the synthesis of phenol-based resins, the vacant ortho positions of the aromatic rings in lignin can react with FA to yield polymeric gels. Multiple examples of utilizing different types of lignin have been described in the literature for preparing gels, that, depending on the drying process, will yield aerogels [20], xerogels [16], or cryogels [17]. The evaporation of liquids from gels at normal atmospheric conditions results in xerogels with a higher density. To obtain lighter materials, sublimation of a frozen solvent via lyophilization results in cryogels. The most effective technique for preserving the porous structure utilizes the extraction of liquid at supercritical conditions where, typically, prior to the process, the solvent is exchanged with CO_2_ [21]. However, supercritical CO_2_ extraction is technically demanding, and despite the widespread use of this method, simpler and more economical methods for material drying are actively researched.

While phenolic resins and resorcinol aerogels are mainly considered as a useful source of porous carbon material, these materials do have some additional distinct advantages—a high carbon yield and high microporosity of the carbonized fibers without activation [22,23,24]. The latter is true also for lignin-derived materials [18,25].

The structure of lignin is highly dependent on its isolation method. The present study concentrated on the organosolv process for lignin extraction, utilizing two different solvent systems. There are not very many studies on using organosolv lignins for the preparation of gels and respective aerogels. Furthermore, the availability of structural information on this type of lignin remains scarce. The current study focused on the preparation and thorough parametrization of organosolv lignins from different types of lignocellulosic biomass as well as their utilization for the preparation of aerogels in different ratios to phenolic compounds. A thorough structural characterization of the obtained aerogels and their properties was carried out.

## 2. Materials and Methods

### 2.1. Chemicals

Ethanol, dioxane, hydrochloric acid, acetic acid, sodium hydroxide, tetrahydrofuran, sulfuric acid, and chloroform-d were purchased from Sigma-Aldrich (Taufkirchen, Germany). All the chemicals were of analytical grade and were used as received. Deionized water from a Milli-Q water purification system (Millipore S. A., Molsheim, France) was used throughout the study. 5-MR with a reported purity of >99% was provided by AS VKG (Kohtla-Järve, Estonia).

### 2.2. Raw Materials and Methodology

Aspen wood chips were provided by Estonian Cell AS, longitudinally sawn pine timber sawdust was provided by Prof. Jaan Kers (Tallinn University of Technology, Tallinn, Estonia), and barley straw was provided by Prof. Timo Kikas (Estonian University of Life Sciences, Tartu, Estonia). All feedstocks were dried in a convection oven at 50 °C up to 8% moisture, followed by grinding to a fine powder, and stored in plastic bags at room temperature.

Organic solvent extraction under reflux conditions was used for the isolation of lignin from biomass and for that, ethanol and dioxane were used. The obtained lignin was further characterized for purity and structural properties by different analytical methods and used for the preparation of lignin-based aerogels, followed by their characterization.

### 2.3. Organosolv Extraction of Lignin

Ground and dried biomass (50 g) was refluxed with 1.5 L of solvent in a 2.0 L round bottom flask with a mechanical stirrer and a reflux condenser for 6 h. The solvent mixture contained 0.28M HCl and 5% *v*/*v* water in ethanol or 0.28M HCl and 2% *v*/*v* water in dioxane. The mixture was filtered through Whatman filter paper, and the solids were washed three times with 50 mL of ethanol or dioxane, depending on the extraction solvent. The combined filtrate and washes were concentrated to about 100 mL by rotor evaporation. Lignin recovery from the pretreatment liquor was performed by precipitation with ultrapure water. For this purpose, the pretreatment liquor was dissolved in 100 mL of acetone and introduced into 2 L of vigorously stirred cold ultrapure water to reduce the solubility of lignin. The mixture was stirred for 60 min, and then the precipitated lignin was separated by centrifugation at 4200 rpm. The recovered lignin was washed with 1 L of ultrapure water three times, centrifuged, dried in a convection oven at 30 °C for 24 h, weighed, and stored for further analysis and use. The lignin extraction yields were calculated according to the following equation:(1)$$\%\;Lignin\;yield=\frac{W_{Extracted\;lignin}}{W_{Lignin\;in\;biomass}}\times 100,$$

### 2.4. Characterization of Lignin

#### 2.4.1. Moisture and Ash Content

For moisture analysis, 0.2 g of organosolv lignin was introduced into the moisture analyzer (Ohaus, Parsippany, NJ, USA) and heated at 105 °C until a constant weight. The ash content was determined by keeping of 0.2 g of the dry lignin in muffle furnace at 550 °C for 4 h and recording the weight of ash [26].

#### 2.4.2. Determination of Lignin and Carbohydrate Content

The analysis of Klason lignin and carbohydrate content involved a two-stage acid hydrolysis of the lignin samples to solubilize and hydrolyze the carbohydrates that were present in the extracted lignin. For that, 200 mg of dry lignin was dissolved in 3 mL of 72% sulfuric acid. The mixture was incubated in a water bath at 30 °C for 1 h. Next, 72 mL of ultrapure water was added to the mixture, and the mixture was autoclaved at 121 °C for 60 min. The solution that was obtained was filtered while hot. The precipitate that retained in the filter was acid insoluble lignin (AIL), which was washed with water, followed by drying in a hot air oven, and weighing. Some amount of lignin was dissolved in the acidic solution (filtrate). That portion of lignin was quantified spectrophotometrically by recording the absorbance of the filtrate at 205 nm. The filtrate was analyzed by a UV–visible spectrophotometer (Cary 50 Bio, Varian, Palo Alto, CA, USA) and the acid soluble lignin (*ASL*) was quantified with the following formula of the Biorefinery test method L2:2016:(2)$$ASL=\frac{A*D*V}{a*b*M},$$

Different notations are: *A* = absorption at 205 nm, *D* = dilution factor, *V* = volume of filtrate, L; *a* = extinction coefficient of lignin, g/L (110 g/L as per TAPPI UM 250), *b* = cuvette path length, cm; *M* = weight of dry sample (in g).

The carbohydrate content of the filtrate that was obtained from lignin hydrolysis was analyzed by an optimized capillary electrophoresis (CE) method. The direct UV-detection of carbohydrates at the 270 nm is possible due to in-capillary reaction of the carbohydrates in the alkali medium, which leads to the formation of UV-absorbing carbohydrate enediolate anion [27,28]. The alkaline electrolyte solution was made up of 136 mM sodium hydroxide and 46 mM disodium hydrogen phosphate dihydrate. The filtrate samples were neutralized with calcium carbonate and injected to CE instrument (7100 CE system, Agilent Technologies, Santa Clara, CA, USA) with a pressure of 35 mbar for 10 s. The applied separation voltage was 19.4 kV. The CE separation was carried out in uncoated fused-silica capillaries of 50 µm id and length 70/80 cm (effective length/total length). Both the capillary and samples were held at 17 °C. New capillaries (Molex, Polymicro Technologies, Phoenix, AZ, USA) were conditioned by rinsing with 1 M sodium hydroxide (30 min), ultrapure water (5 min), and the electrolyte solution (5 min). Between analyses, the capillaries were flushed with 5% *v*/*v* acetic acid for 3 min, with 1 M sodium hydroxide for 3 min, with ultrapure water for 3 min, and finally with electrolyte solution for 5 min.

A total of eight sugars raffinose, fucose, cellobiose, galactose, glucose, mannose, arabinose, xylose, and 5-hydroxymethylfurfural were quantified in the linear range of 0.1–4.0% dw with determination coefficients around 0.99.

#### 2.4.3. FTIR Spectroscopy

The infrared (IR) absorption spectra of lignins were obtained using a FTIR spectrometer IRTracer-100 (Shimadzu, Kyoto, Japan). A spectrum of the blank KBr tablet and the scan of the KBr tablet with test substance (100:1) were recorded in the range 400–4000 cm^−1^ with a resolution of 4 cm^−1^. The IR absorption spectrum of the lignin was obtained by subtraction of the test substance scan with the scan of the blank. A total of 32 spectra were averaged per sample and analyzed using the Lab Solutions software (Shimadzu, Kyoto, Japan).

#### 2.4.4. NMR Characterization

For the sample preparation, about 35.0 mg of lignin was dissolved in 600 μL of DMSO-d_6_. The full assignment of 2D-Heteronuclear Single Quantum Coherence (HSQC) was based on the spectra that were obtained from a 500 MHz Agilent DD2 spectrometer that was equipped with an inverse detection probe. HSQC experiment was recorded at 25 °C by applying the standard multiplicity edited HSQC pulse sequence with adiabatic pulses in ^13^C channel. The 2D data were acquired in 256 increments (64 scans/increment) in f1 dimension with 20,000 Hz spectral width. The acquisition of the f2-dimension was zero filled to 2000 points, while the f1 dimension was linear predicted to 512 points, followed by zero filling to 1000 points. Prior to Fourier transformation, a gaussian window function was applied in both dimensions. The resulting free induction decays were processed using MestReNova software. The HSQC cross signals were assigned by comparing with the literature [29,30,31,32,33,34].

#### 2.4.5. ^31^P-NMR Quantitative Analysis

Free hydroxyl group concentrations of the six organosolv lignin samples were estimated by using ^31^P NMR, based on the derivatized OH groups with an appropriate phosphitylation reagent. Sample preparation for ^31^P NMR analysis was carried out according to the procedure from Balakshin et al. [15]. Particularly, organosolv lignin (about 25.0 mg) was weighed into an Eppendorf tube, followed by adding 500.0 µL of pyridine/CDCl_3_ with ratio 1.6:1 (solution I). Then, 100.0 µL of internal standard (IS) (20.0 mg/mL) that was prepared in solution I, was added. Subsequently, approximately 5.0 mg/mL of chromnium (III) acetylacetonate as a relaxation agent was pipetted into the same tube. Finally, the mixture was derivatized with 70.0 µL of 2-chloro-4,4,5,5-tetramethyl-1,3,2-dioxaphospholane (TMDP). The tube was closed tightly and manually shaken to obtain a homogeneous mixture. About 600 µL of the phosphitylated lignin was transferred into an NMR tube and analyzed. After phosphitylation (Figure 2), the signals of the hydroxyl groups from aliphatic, syringyl, guaiacyl, *p*-hydroxyphenyl, and carboxylic acid were identified from the NMR spectra. Their quantification was based on the peak integral of the targeted OH group relative to an IS endo-N-hydroxy-5-norbornene-2,3-dicarboximide. The data were obtained on a Bruker 400 MHz spectrometer that was equipped with 5 mm BBO BB-1H/D probe. The ^31^P spectra were acquired at 25 °C with an inverse gated decoupling pulse sequence (zgig) with 768 number of scans, 10s for relaxation delay. The data were then processed by using Topspin 4.0.8.

All the ^31^P spectrum were calibrated by the signal from IS at 151.8 ppm; and the integration as well as assignment were based on the study by Balakshin et al. [15].

#### 2.4.6. Lignin Molecular Weight Distribution

As a routine, the weight-averaged molecular weight (M_W_) and weight distributions (M_W_D) are determined by size exclusion chromatography (SEC), where the M_W_ data are related to calibration standards of known M_W_. The lignin samples were dissolved in tetrahydrofuran (THF, 1 mg/mL) and analyzed on a Prominence LC-20A Modular HPLC System (Shimadzu, Japan) that was equipped with a photodiode array detector. All lignins that were prepared in this study were soluble in THF at 1 mg/mL. The columns that were used were a series of two 300 mm × 7.5 mm i.d., 3 μm, MesoPore, with a 50 mm × 7.5 mm i.d. guard column of the same material (Agilent Technologies). The samples were eluted at 40 °C with 1 mL/min THF stabilized with 250 ppm butylated hydroxytoluene (BHT) and detected at 254 nm. The system was calibrated using polystyrene GPC/SEC calibration standards (EasiVial PS-L, Agilent Technologies) in the range of molecular weight at peak top (Mp) 46 380−162 Da. Mp, number-averaged molecular weight (Mn), M_W_, and polydispersity (M_W_/Mn) were determined by the Lab Solutions software (Shimadzu).

### 2.5. Preparation and Characterization of Lignin-Based Aerogels

The organosolv lignins that were obtained (Table 1) were used to prepare the lignin-5-methylresorcinol-formaldehyde (L-5-MR-FA) aerogels. Aerogels with partly replaced 5-MR with lignin were prepared according to previously published protocols [14,35], which were combined and improved as needed.

Firstly, the lignins were homogenized in ultrapure water and stirred at 85 °C for 90 min with 0.09 % (wt. % based on lignin) of NaOH, pH of mixture was close to 10. After that, a water solution of 5-MR and FA, prepared according to the mass ratios: (5-MR+L)/FA = 1.25; 5-MR/L = 1/3 (75% 5-MR is replaced by lignin); water content equal to 80%, was added to the cooled mixture of lignin and homogenized by mixing on a vortex. The mixture was kept at 85 °C to allow gelation, where the time of the gelation was determined visually. The gels were kept at room temperature for curing in 1% of acetic acid for 12 h, and then placed in acetone, which was changed every 24 h for four days. After the solvent exchange step, the gels were dried using supercritical CO_2_, dynamic drying (100 bar 1.5 h 25 °C, 120 bar 2.5 h 45 °C).

The surface morphology of the L-5-MR-FA aerogels were examined with a high resolution scanning electron microscope (SEM) Zeiss EVO MA 15 SEM at an accelerating voltage of 10 kV. For imaging, pieces of aerogel samples were fractured into smaller parts to open the internal structure and view the particles close morphology. The fractured pieces were attached with double adhesive tape to the stub and coated with the Ag/Pd conductive layer in the Fine Coat Ion Sputter JFC-1100.

The pore structures of the aerogel samples were investigated by using the N_2_ adsorption-desorption method at 77 K with Quantachrome Autosorb iQ apparatus in the relative pressure range of 0.005 to 0.995. Prior to analysis, aerogel powder was degassed at 105 °C for 24 h to remove surface impurities and moisture. The N_2_ adsorption-desorption method at 77 K was used in the relative pressure P/P_0_ range of 0.025 to 0.995. The specific surface area (S_BET_) was calculated by applying the BET (Brunauer-Emmett-Teller) method and the total pore volume was determined by the volume of N_2_ that was adsorbed at relative pressure P/P_0_ = 0.99. Pore size distribution was determined using density functional theory (DFT).

Ground samples of aerogels were spotted on a diamond crystal and analyzed on an IRTracer-100 FTIR spectrophotometer (Shimadzu, Kyoto, Japan) in attenuated total reflection (ATR) mode. The spectra were recorded over 400–4000 cm^−1^ range by averaging 20 scans at a maximum resolution of 2 cm^−1^ and analyzed using Lab Solutions software (Thermo Scientific, Waltham, MA, USA).

## 3. Results and Discussion

### 3.1. Lignin Structural Characterization

#### 3.1.1. Purity and SEC Analysis

A total of three types of lignocellulosic biomass were organosolv-pretreated with two different solvents (ethanol and 1,4-dioxane) under reflux conditions as mentioned in the experimental section. Aspen, pinewood, and barley straw biomass had an initial lignin content ca. 27, 33, and 21% per dry weight, respectively. As can be seen in Table 1, the overall lignin recovery yield is around 20% for ethanol extraction. This low yield could be partially attributed to the formation of colloidal suspensions of lignin in the water washes that would not clear even after prolonged centrifugation. Also, for the ethanol solvent system, the lower yield is probably due to the lower extraction temperature (82–85 °C) compared to the dioxane systems (~95 °C) that led to less lignin fragmentation and less lignin solvolysis and therefore less overall lignin yield. The higher temperature and better solubilization ability of dioxane resulted in a maximum lignin yield of 61% for untreated pine wood.

The purity of the obtained lignins was evaluated by the ash, sugar, and Klason lignin contents. The lignin was treated with sulfuric acid to obtain Klason lignin content and release sugars that were present in the lignin. The acid solution was then filtered and neutralized, and the filtrate was analyzed by capillary electrophoresis (CE). Klason lignin contents varied between 95–100% (Table 1). The lowest value was observed for the lignin of pine due to the increased proportion of residual carbohydrates (0.3–0.6%), mainly mannose. Differences in the composition of carbohydrate impurities suggest that lignin-carbohydrate complexes in softwoods [36] are structurally distinct when compared to hardwoods and grassy biomass. The results show that no sugar was released from aspen and barley straw lignin after acid treatment. The ash contents for all lignins were insignificant (below 0,1% dw).

SEC was used to determine the M_W_D and polydispersity index (PI) of the obtained lignin. The Mn, weight average molecular weight (M_w_), and polydispersity (M_w_/Mn) of ethanol organosolv (EOL) and dioxane organosolv (DOL) lignins were calculated from SEC chromatograms. The results are shown in Table 1. For lignins that were isolated with ethanol, slightly higher values of M_W_ and Mn were observed when compared with lignin samples that were isolated with dioxane. Notably, EOL and DOL lignins that were obtained from different biomasses had similarly narrow polydispersity of 1.4–1.5, indicating homogeneity of the polymeric material.

A peak at longer retention time (around 1060 s) was present in the EOL-barley straw lignin and was remarkably abundant in the EOL-pine lignin (data not shown). This peak had a calculated molecular weight ≈ 0.2 kDa, was absent in all DOL-lignins. It was shown by Faleva et al. that a high ethanol concentration could favor the formation of ethylesterified compounds that were derived from either lignin or hemicellulose [29]. Due to their low water solubility, these derivatives could coprecipitate together with the larger lignin compounds.

#### 3.1.2. FTIR Analysis

The IR absorption spectra of the six lignins that were studied were recorded in the 400–4000 cm^−1^ region (Figure 3). Absorption bands corresponding to bond vibrations in the FT-IR spectra were assigned based on previously reported data [37,38]. The IR spectral profiles are similar in all the isolates, which indicates that the “core” of the structure of organosolv lignins is similar for different kinds of biomass. The band at 3428 cm^−1^, which is attributed to O-H stretching absorption due to the phenolic and aliphatic hydroxyl groups, had a similar absorption intensity in all lignins. However, the relative intensities of the absorption bands at 2938 and 2849 cm^−1^, assigned to C-H stretching vibrations in the methyl and methylene groups were quite different. These bands, which are mainly attributed to methoxy groups, were substantially higher for the EOL lignins and presented relatively lower absorbance bands for DOL lignins. The carbonyl stretching vibration at 1710 cm^−1^, which is caused by stretching of unconjugated ketones, conjugated aldehydes, and carboxylic acids aromatic ring appeared in all spectra with different intensity. The bands at 1595 and 1516 cm^−1^ are caused by aromatic skeletal C=C vibrations, while the band at 1464 cm^−1^ corresponds to asymmetrical C-H deformations in methyl (-CH_3_) and methylene (-CH_2_-) groups. The band at 1424 cm^−1^, which also corresponds to aromatic skeletal vibrations (C=C) (guaiacyl-syringyl) combined with C-H in-plane deformation, has lowest intensity in barley straw lignin spectra, and the band at 1330 cm^−1^ corresponding to C-O stretching (S) was clearly observed only on the aspen (hard wood) lignin spectra. The bands between 1300 and 1000 cm^−1^ are attributed to stretching of the C-O ether or ester linkage. Aliphatic ethers give a strong asymmetric stretch at 1122 cm^−1^ in aspen OL, while for the barley straw lignin, its intensity is negligible.

FTIR spectroscopy showed that the lignins under investigation had some structural differences that were caused by the used extraction solvent as well as biomass source, which will be covered in further detail in this study.

#### 3.1.3. ^31^P-NMR Quantitative Analysis

In the present study, the results only assigned for syringyl OH instead of combining its peak area with condensed phenolic units in lignin. Therefore, the additional aromatic OH region was taken into account from 144.0 to 137.0 (ppm) containing not only OH from H-/G-/S-units but also from the condensed units in lignin. The contribution of signals from specific hydroxyl groups were integrated and calculated quantitatively on the basis of the IS (Table 2). In general, the distribution of hydroxyl content from aliphatic chains and *p*-hydroxyphenyl groups were higher when using ethanol to extract lignin from all types of biomass. In contrast, 1,4-dioxane gave better extraction yield as evidenced by a significant amount of phenolic compounds, including OH groups from the three main components in lignin H-, G- (uncondensed/condensed unit), and S-units.

The amounts of syringyl OH were found significantly higher in organosolv lignin from hardwood (i.e., aspen) than from softwood (i.e., pine). The highest amount was attributed to DOL-Aspen with 1.32 (mmol OH g^−1^ of dried lignin). Approximately one tenth of the amount of phenolic hydroxyls were attributed to the *p*-hydroxyphenyl functional group that was present in DOL-Aspen (0.12 mmol OH g^−1^ of dried lignin) compared to the OH groups from S-units. By contrast, phenolic OH groups from G moieties in organosolv lignin from pine were predominant, especially in DOL-Pine with 1.35 (mmol OH g^−1^ of lignin) while phenolic OH groups from the S-units were not detected. Barley straw was classified as a member of the grassy biomass family, therefore, its ^31^P NMR showed all three phenolic OH groups in which the content from G-OH was slightly higher than the other two. The results showed a strong agreement with a previous study on lignin composition that described the monolignol distributions in different biomasses– S/G ratio is greater in hardwoods than softwood, while the lignins from grassy biomass contain all three components (H, G, S) [39].

It is clear that H- and G-units are able to react with FA to synthesize the lignin-based resins while the S (no active site is present) unit is not. This was seen in Table 2 where softwood and grass showed higher potential for forming good aerogels because of a higher content of G- and H-units.

#### 3.1.4. 2D HSQC Characterization

In addition to the results that were obtained from phosphorous NMR, six organosolv lignin samples were further structurally characterized by 2D Heteronuclear Single Quantum Coherence (HSQC) NMR spectroscopy. HSQC is a powerful technique that shows the correlation of hydrogen atoms that are bonded directly to a carbon atoms, giving detailed information about inter-unit linkages in lignin [40,41]. In this study, HSQC was only employed for qualitative analysis to acquire more information about lignin substructures.

It was reported that the region of aliphatic side chain (δ_C_/δ_H_ 50–10/3.0–1.1 ppm) was not informative compared to others in terms of lignin linkages [31]. Therefore, only two regions from the HSQC spectrum were considered: (i) oxygenated aliphatic side chain region (δ_C_/δ_H_ 100–40/6.0–2.5 ppm), and (ii) aromatic/unsaturated region (δ_C_/δ_H_ 150–100/8.0–6.0 ppm).
(i)The oxygenated aliphatic side chain region of organosolv lignins and its correlations are shown in Figure 4 and Appendix A, respectively. As expected, the methoxy groups are present in all the lignin spectra. Under acidic hydrolysis condition, the cleavage of α-aryl ether bonds are favored over the cleavage of β-O-4′ bonds, resulting in the formation of intermediate α-carbocation. This active carbon with the presence of a nucleophile source from ethanol would lead to the substitution at C_α_ [42,43]. As a consequence, a strong signal for α-ethoxylation in the β-O-4′ substructure (A’) can be observed (δ_C_/δ_H_ 63.60/3.32 ppm) in all the ethanol organosolv lignin samples (EOL) [29].

The presence of other dominant substructures, such as β-O-4′ (A), resinol (B), phenylcoumarans (C), and cinnamyl alcohol end-groups (I) were found in all types of EOL, except in lignin from pinewood where resinol fragment was not detectable (Figure 4E). Surprisingly, the cross peak intensity from β-aryl ether bond in A/A’ was significantly reduced in dioxane organosolv lignin (DOL), indicating vigorous cleavage of this bond to release lignin from the lignocellulosic matrix when using dioxane as the solvent. This phenomenon could be explained by the higher boiling point of dioxane (95 °C vs. 85 °C of ethanol) [9], which leads to a higher temperature during the extraction process, assisting in the breakdown of more aryl ether bonds. Additionally, C_α,γ_-H_α,γ_ Hibbert’s ketone (Hk) structures were seen in all the DOL spectrums, while only C_γ_-H_γ_ was observed in EOL. The occurrence of Hk in organosolv lignins must be one of the consequences of using an acid catalyst in the solvolysis process [9]. The low intensity of Hk_α_ in EOL can be explained by the fact that its formation likely competes with the ethoxylation reaction in the excess of ethanol. It has been noted that cinnamyl alcohol end-groups (I) can be observed only in EOL, while β-1′ linkage (E) has only been described for hardwood lignins (Figure 4A,B) [6].
(ii)Figure 5 shows the aromatic/unsaturated region in six lignin samples from different biomasses. Cross peaks from S-units and its oxidized-C_α_ (S’) were more pronounced in the aspen and barley straw, while there was no signal in the spectra from pine. In contrast, the G-unit was present in all the lignins, whereas H was only observed in all EOL-s. These results are in good agreement with the data from the analysis of the hydroxyl contents that were obtained from phosphorous NMR where the functional OH groups from the S-unit were not detected in organosolv lignin from pine.

In most of the cases, Hk-signals were frequently assigned for C_α,γ_-H_α,γ_ in many ways thanks to the past efforts by Miles-Barrett et al. [11] who synthesized an Hk analog from S-/G-units as well as Hibbert’s ketone bound to a lignin structure that resulted from the cleavage under acidic organosolv treatment. As a result, it enabled us to extract more information from the HSQC spectrum. In the studied lignins, aromatics of syringyl/guaiacyl that were bound to Hk (S_2,6_-LBHK, G_2_-LBHK, G_5_-LBHK, G_6_-LBHK) were assigned at δ_C_/δ_H_ 106.67/6.53, 113.31/6.84, 112.89/6.90, and at 121.36/6.64, respectively. Furthermore, other fragments from methyl-substituted phenylcoumaron (P) were successfully confirmed by Faleva et al. [8] and were present in DOL-Aspen/barley straw spectra in our studied cases. It can be seen that barley straw lignin (Figure 5C,D) contains ferulate (Fer) and *p*-coumarate (pCA) substructures as a result of acylation of lignin side chains [5]. Along with Fer and pCA, tricin (T) is well distributed in grasses/straw, therefore their signals were strongly observed and assigned according to a previous study [7]. Interestingly, the presence of *p*-benzoate (PB) was only seen in aspen with the assignment of C_2,6_-H_2,6_ at δ_C_/δ_H_ 131.13/7.69 as reported previously [10].

From the point of view of gel-forming reactions, the most valuable elements are aromatic rings with hydroxyl group in ortho-or para-position-pCA, Fer, T, PB.

### 3.2. Characterization of the Obtained Aerogels

#### 3.2.1. Drying Shrinkage, Bulk Density, and Gelation Analysis

Drying shrinkage, bulk density, and gelation time of AG-5-MR-FA and aerogels, where 75–85% of the 5-MR was replaced by lignin, are presented in Table 3. The drying shrinkages of lignin-based aerogels were higher than 45%, but the bulk densities were still quite low (0.15–0.46 g/cm^3^), compared to dense structured AG-5-MR-FA, which had low shrinkage (37%), but a bulk density of 0.20 g/cm^3^.

The solubility of lignin in water is important because when not fully dissolved, its reactivity is low and could act as aerogel filler, not as one of the precursors. This was the case for EOL-Pine lignin. The gelation time at 85 °C for lignin-based gels was relatively long, up to seven hours for lignin from aspen. While longer gelation times are characteristic for AG-EOLs, then AG-DOL gelation times are more comparable to the gelation times for common 5-MR-FA gels. SEC analysis of the obtained lignins (Table 1) reveals that using dioxane as the solvent for the extraction of lignin gives products with a higher yield and lower molecular weight. This could be explained by the higher temperature boiling point of dioxane compared to ethanol, which can assist the cleavage of aryl-ether bonds in lignin [7]. Furthermore, the results from 2D-HSQC (Figure 4) show that the peak intensities of the corresponding β-O-4′ linkages in lignin decreased significantly when using dioxane, compared to ethanol under the same extraction condition. It can, therefore, be concluded, that the delignification of biomass with dioxane gives smaller fragments of lignin, which increases their solubility and increases the aerogel formation ability. Consequently, AG-DOL show shorter gelation time compared to AG-EOL (Table 3).

In comparison, the shrinkage at the drying state of the material that was obtained from EOLs was higher than the ones from DOLs (except DOL Pine with similarity in their shrinkage). A lesser extent of shrinkage of AG-DOLs was likely due to the higher degree of cross-linkage in the material. The overall higher signal from phenolic groups (Table 2) in DOL gave more possibility to form an aerogel with FA.

The experiments of replacing more than 75% of 5-MR were also performed and these demonstrated that it is possible to replace even up to 85% of the 5-MR with lignin (Table 3). In the experiments where 95% of 5-MR was substituted with lignin showed no gelation of the mixture during 24 h at 85 °C for all the lignins that were used.

#### 3.2.2. Measurement of Porosity and Surface Area

Aerogels are porous materials and the two parameters that characterize this type of material best are porosity and surface area. The common technique that is used for the determination of these parameters is N_2_ adsorption/desorption analysis, which allows the calculation of average pore sizes and distribution as well as the surface area of the material. The N_2_ adsorption-desorption isotherms of the prepared aerogels (lignin-based, where 75% of the 5-MR was replaced by lignin and AG-5-MR-FA) are depicted in Figure 6.

According to International Union of Pure and Applied Chemistry (IUPAC) classification, the obtained N_2_ adsorption/desorption isotherms for aerogels are characteristic of adsorbents that are mesoporous (types IV and V). The adsorption hysteresis is clearly seen and can be utilized to find correlations between the shape of the hysteresis loop and the texture (e.g., pore size distribution, pore geometry, and connectivity) of a mesoporous material.

The initial region of the isotherms at low P/P_0_, which indicates the presence of micropores in addition to mesopores in the gel structure, is very similar to all samples and not characteristic for these aerogels specifically. It is well-known that organic resorcinol-FA aerogels do not show microporosity in their structure. Hysteresis loops for gels in the P/P_0_ range 0.2–1.0 show that mesoporosity is dominant in their structures.

The hysteresis loop of 5-MR-FA is of type H1, coinciding with the network structure of agglomerates of uniform spherical particles. The addition of lignin into the structure of aerogel tends to change the loop shape from type H1 to H2, suggesting that the primary particle shape is not well defined and some disordering with a partial opening of the network structure has taken place.

According to IUPAC, pores with diameters larger than 50 nm should be called macropores, while pores with diameters in the range of 2–50 nm should be called mesopores, and pores with diameters smaller than 2 nm should be called micropores [44].

All of the aerogels that were obtained during the study contained mainly mesopores with an average pore diameter in the range of 8.90–13.97 nm, while the 5-MR-FA aerogel, which does not contain lignin, had average pore volume of 30.60 nm. The pore size distributions of the prepared aerogels are depicted in Appendix A. The specific surface area (S_BET_)values were in the range of 271.3–452.9 m^2^/g, which are even higher, compared to the aerogel, which does not contain lignin (224.3 m^2^/g). The specific surface area, total pore volume, average pore diameter of lignin-based (75% of the 5-MR was replaced by lignin) and 5-MR-FA aerogels are presented in Table 4. We found no correlation between the average pore volumes or S_BET_, and the extraction solvent, but the solubility of EOLs was always lower than DOLs (both pretreated with NaOH). However, there were similarities of aspen- and barley-based lignin aerogels, where the S_BET_ values were considerably higher for DOLs. For AG-EOL Pine, this might be due to the fact that EOL Pine was not completely solubilized and there was even a small amount of precipitation, which meant that the bottom layer of the aerogel was darker and not homogeneous. For the characterization experiments, this insoluble bottom layer was removed from the samples.

#### 3.2.3. FTIR Analysis

The Fourier transform infrared spectroscopy technique is sensitive to differences in the structure of lignin-based and AG-5-MR-FA aerogels. AG-5-MR-FA and all the aerogels, where 75% of the 5-MR was replaced by lignin were investigated by ATR FTIR spectroscopy (Figure 7).

The main bands that were located at 1589, 1455, and 1110 cm^−1^ have been described in the literature for resorcinol-FA aerogels that were studied by FTIR [45], which were confirmed by the AG-5-MR-FA spectra shown in Figure 7. The band around 1455 cm^−1^ is associated with the formation of methylene bridges between the aromatic rings as described in the literature earlier. The main band in the region of 1589 cm^−1^ is due to the C-C stretching in the aromatic rings, while the bands at around 1110 cm^−1^ can also be assigned to the C-OH stretching and deformation in phenolic groups. These bands are seen in both lignin-based and 5-MR-FA aerogels.

The main differences are observed in the region around 2900 to 2950 cm^−1^, where the band intensities are increased from CH stretching in the aromatic methoxy groups and in the methyl and methylene groups of side chains. They are intensified in lignin spectra but less seen in lignin aerogels.

Also, in the FTIR spectra of lignin-based aerogels, there is an observable peak at 1507.40 cm^−1^, which is a characteristic peak for lignin and does not exist in AG-5-MR-FA spectra. Another proof of the participation of lignin in polycondensation reactions leading to gelation is the decreasing intensity of bands corresponding to methoxy groups and primary alcohols, whereas bands that are related to C=O bond are still noticeable in the spectra of lignin-based aerogels in comparison with AG-5-MR-FA [46].

#### 3.2.4. SEM Analysis

SEM images of the obtained lignin-based aerogel samples, where 75% of the 5-MR was replaced by lignin, are similar with characteristic features of 5-MR-FA aerogel, with particle sizes of 10–50 nm and uniformly distributed pores. However, a porosity assessment with SEM showed the differences from 5-MR-FA aerogels, which confirmed the N_2_ adsorption/desorption analysis results. The morphology of the aerogels that were obtained during the study are presented in Figure 8.

## 4. Conclusions

It is possible to produce FA aerogels from lignin with the small addition of 5-MR (25% to 15%) using aspen, pine, and barley lignins that were obtained by organosolv extraction. It shows that the organosolv lignin extraction process gives material that has good reactivity to form gels with FA resulting in monolithic mesoporous aerogels with relatively high surface area –from 350 to 450 m^2^/g. The primary particle shape in aerogels is not uniformly spherical but loosely defined and somewhat disordered with a partial opening of the network structure when compared to 5-MR-FA aerogels. Pretreatment of the precursor lignin with NaOH allows increased the solubility of the lignin and its further use in the base catalyzed synthesis of 5-MR-FA-Lignin aerogels. The obtained lignin-based organic aerogels will be further studied as adsorbents and raw material for porous carbon aerogels.

## Figures and Tables

**Figure 1 materials-15-02861-f001:**
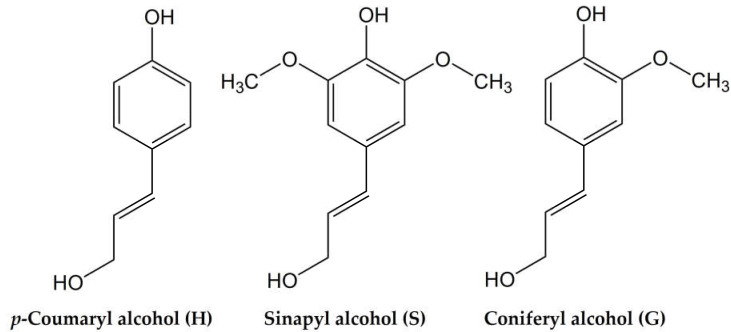
Structures of lignin monolignol units.

**Figure 2 materials-15-02861-f002:**
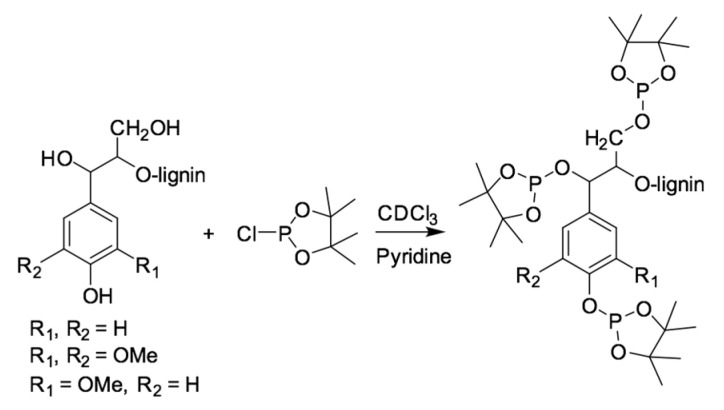
Phosphitylation of hydroxyl groups in lignin with TMDP reagent.

**Figure 3 materials-15-02861-f003:**
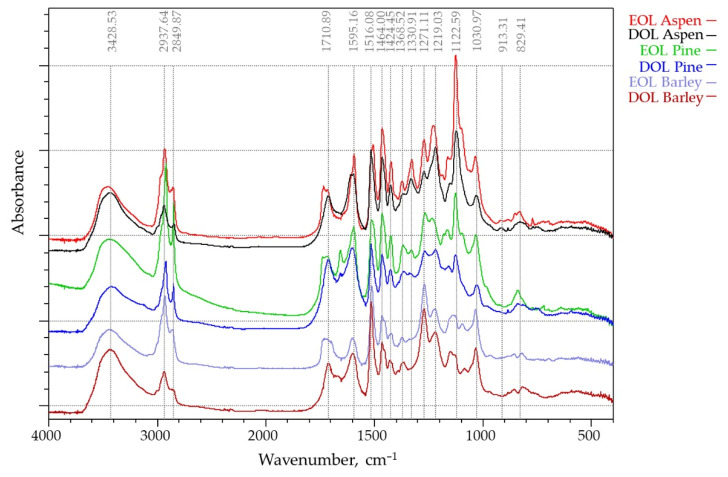
FT-IR spectra of EOL and DOL lignins that were isolated from three different biomass feedstocks (aspen, pine, and barley straw).

**Figure 4 materials-15-02861-f004:**
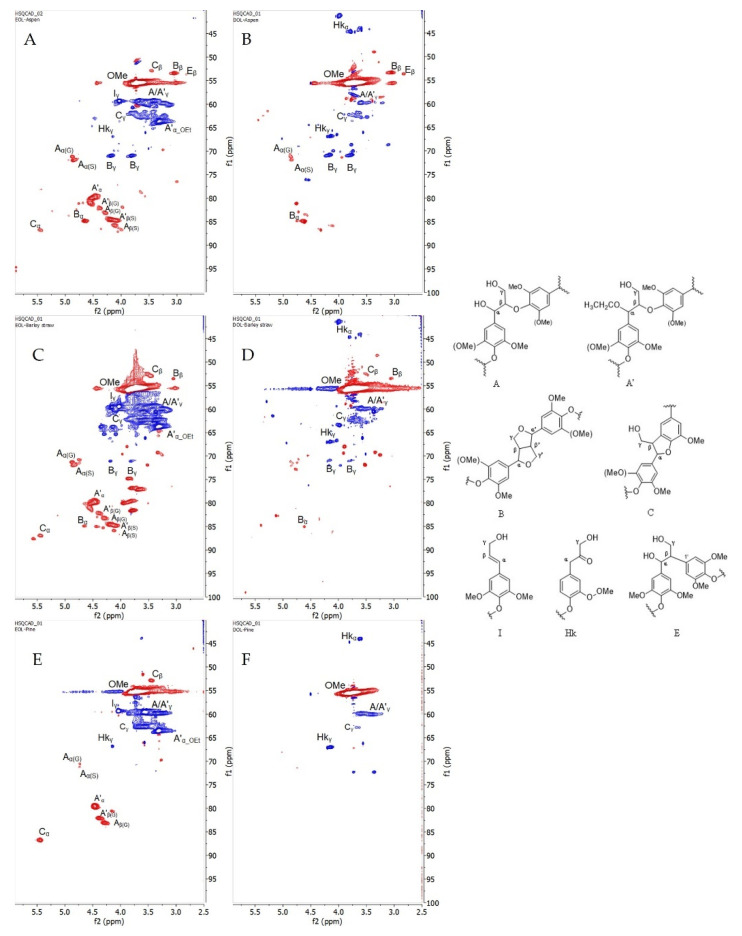
(**Left**) HSQC spectra that were presented in the oxygenated aliphatic side chain region from six organosolv lignins, (**A**) EOL-Aspen, (**B**) DOL-Aspen, (**C**) EOL-Barley straw, (**D**) DOL-Barley straw, (**E**) EOL-Pine, and (**F**) DOL-Pine. Red cross peaks are CH_3_ and CH; blue cross peaks are CH_2_. (**Right**) (**A**) β-O-4′ alkyl-aryl ethers; (**A’**) β-O-4′ alkyl-aryl ethers with ethoxylated α-OH; (**B**) resinols; (**C**) phenylcoumarans; (**I**) cinnamyl alcohol-end groups; (**Hk**) Hibbert’s ketone (end-group); (**E**) β-1′ linkage.

**Figure 5 materials-15-02861-f005:**
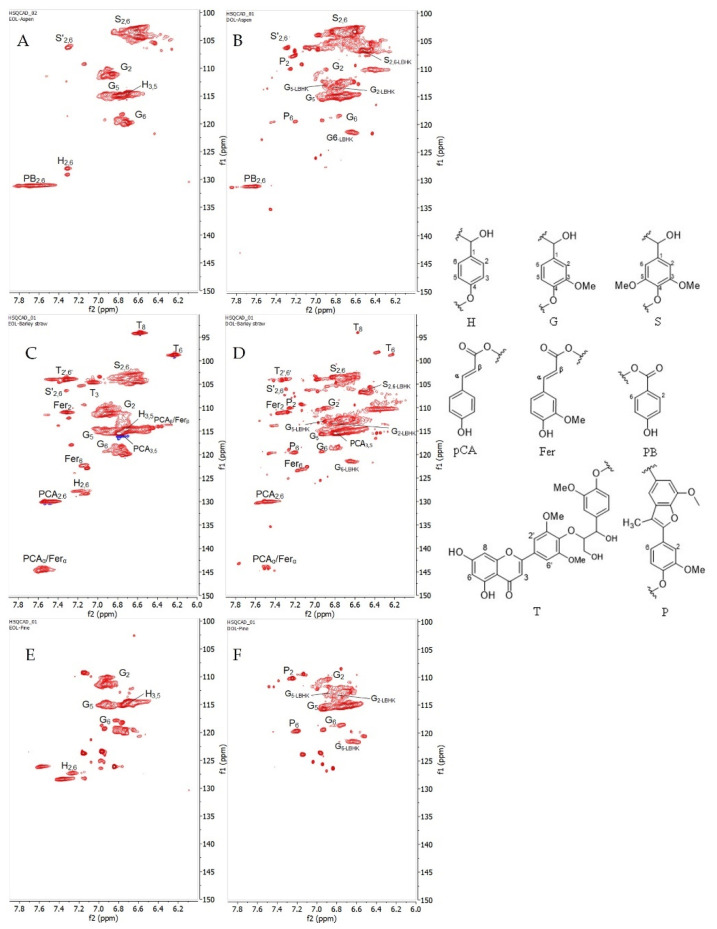
(Left) HSQC spectra that were presented in aromatic/unsaturated region from six organosolv lignins, (**A**) EOL-Aspen, (**B**) DOL-Aspen, (**C**) EOL-Barley straw, (**D**) DOL-Barley straw, (**E**) EOL-Pine, and (**F**) DOL-Pine. Red cross peaks are CH_3_ and CH. (Right) (**H**) *p*-hydroxyphenyl unit; (**G**) guaiacyl unit; (**S**) syringyl unit; (**pCA**) *p*-coumarates; (**Fer**) ferulates; (**PB**) *p*-benzoate; (**T**) tricin; (**P**) methyl-substituted phenylcoumaran.

**Figure 6 materials-15-02861-f006:**
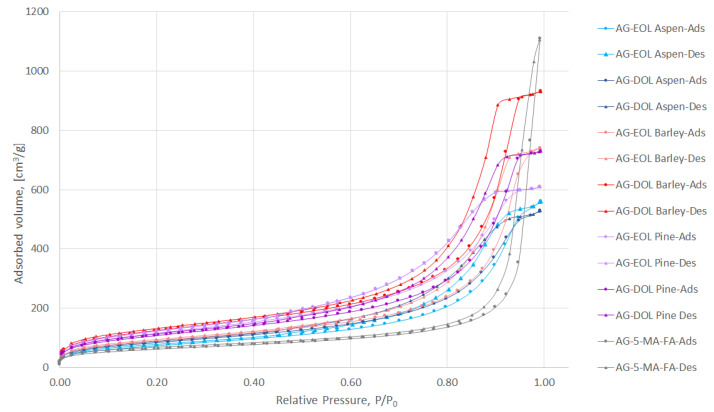
N_2_ adsorption/desorption isotherms of lignin-based (75% of the 5-MR was replaced by lignin) and 5-MR-FA aerogels.

**Figure 7 materials-15-02861-f007:**
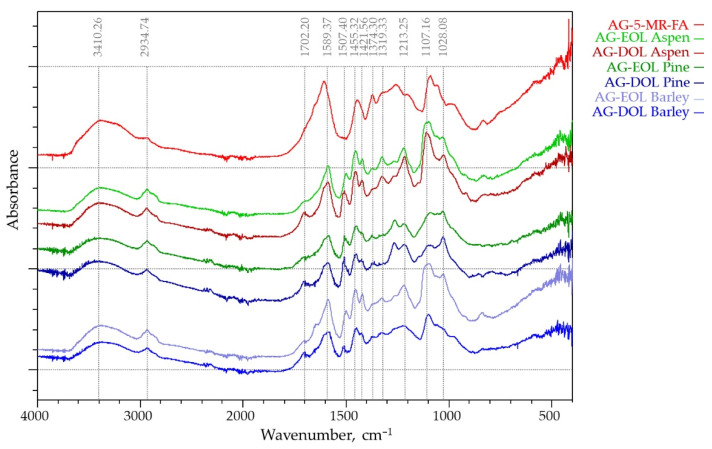
ATR FTIR spectrums of lignin-based (75% of the 5-MR was replaced by lignin) and AG-5-MR-FA aerogels.

**Figure 8 materials-15-02861-f008:**
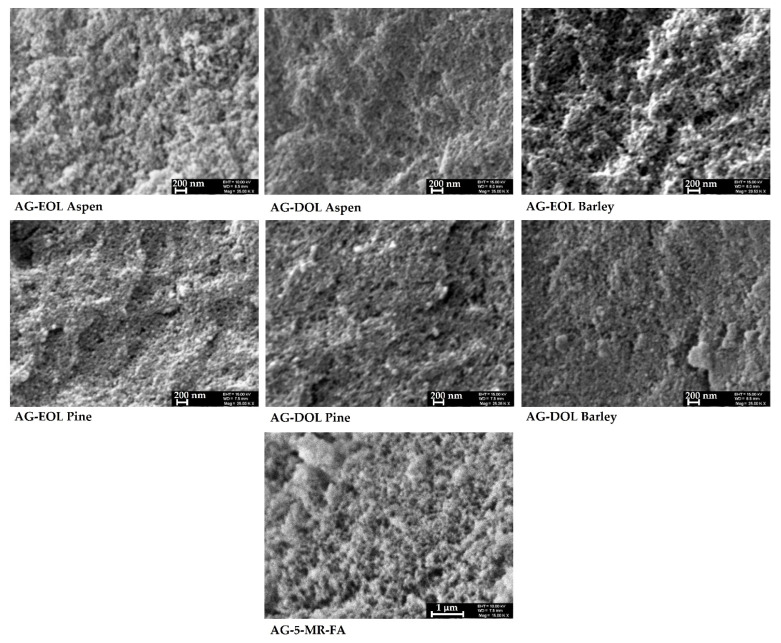
Examples of the SEM images of lignin-based (75% of the 5-MR was replaced by lignin) and 5-MR-FA aerogels.

**Table 1 materials-15-02861-t001:** Physicochemical characterization of organosolv lignin that was extracted from different sources of biomass with two different solvents (ethanol organosolv lignin- EOL and dioxane organosolv lignin—DOL).

Lignin ID	Biomass Source	Solvent Used	Lignin Yield %	Klason Lignin * %	Total Sugars %	M_W_	Mn	PI
EOL-Aspen	Aspen wood	Ethanol	15 ± 2	95 ± 5	nd	2601	3743	1.4
DOL-Aspen	Aspen wood	Dioxane	44 ± 6	104 ± 5	nd	1925	2670	1.4
EOL-Barley straw	Barley straw	Ethanol	24 ± 3	96 ± 5	nd	2187	3211	1.5
DOL-Barley straw	Barley straw	Dioxane	49 ± 7	102 ± 5	nd	1809	2469	1.4
EOL-Pine	Pine wood	Ethanol	21 ± 3	89 ± 4	0.27 ± 0.05	2343	3391	1.4
DOL-Pine	Pine wood	Dioxane	61 ± 9	101 ± 5	0.57 ± 0.09	1959	3450	1.8

nd–not detected (content of each of eight sugars is below LOD 0.1% dw). ± expanded uncertainty (95% confidence interval). * Klason lignin includes both acid soluble and acid insoluble.

**Table 2 materials-15-02861-t002:** Six organosolv lignin samples and their hydroxyl contents (mmol OH/g of dried lignin) that were determined by ^31^P NMR according to their integration for specific type of OH group.

Chemical Shift (ppm)	Types of OH	EOL-Aspen	EOL-Barley Straw	DOL-Aspen	DOL-Barley Straw	EOL-Pine	DOL-Pine
149.0–145.5	Aliphatic OH	2.38	2.08	1.50	0.94	1.90	1.61
143.5–142.0	Syringyl OH	0.35	0.11	1.32	0.53	0	0
140.2–139.0	Guaiacyl OH	0.47	0.48	0.55	0.62	0.73	1.35
138.0–137.5	*p*-Hydroxyphenyl	0.23	0.23	0.12	0.16	0.08	0.07
134.7–134.5	Carboxylic acid	0.05	0.02	0.20	0.25	0.04	0.15
144.0–137.0	Aromatic OH	1.14	0.92	2.49	1.76	0.97	2.59

**Table 3 materials-15-02861-t003:** The drying shrinkage, bulk density, and gelation time of lignin- based and 5-MR-FA aerogels.

Sample ID	AG-EOL Aspen		AG-DOL Aspen		AG-EOL Barley		AG-DOL Barley		AG-EOL Pine		AG-DOL Pine		AG-5-MR-FA
Amount of 5-MR replaced by lignin, %	75	85	75	85	75	85	75	85	75	85	75	85	0
Drying shrinkage, %	77.7	45.9	61.8	73.9	70.9	72.6	61.4	50.4	63.4	47.1	65.1	51.8	37.0
Bulk density, g/cm^3^	0.36	0.15	0.24	0.28	0.33	0.46	0.35	0.19	0.28	0.18	0.33	0.20	0.20
Gelation time, h at (85 ºC)	7	13.5	2	2	4	18	1.5	10	2	3.5	1	3.5	<1

**Table 4 materials-15-02861-t004:** The specific surface area, total pore volume, and average pore diameter of lignin-based (75% of the 5-MR was replaced by lignin) and 5-MR-FA aerogels.

Sample ID	BET Surface Area, m^2^/g	Total Pore Volume, cm^3^/g	Average Pore Diameter, nm
AG-EOL Aspen	271.3	0.87	12.81
AG-DOL Aspen	304.7	0.82	10.73
AG-EOL Pine	424.1	0.94	8.90
AG-DOL Pine	393.2	1.13	11.47
AG-EOL Barley	328.1	1.15	13.97
AG-DOL Barley	452.9	1.44	12.73
AG-5-MR-FA	224.3	1.72	30.60

## Data Availability

The data that are presented in this study are available upon request from the corresponding author.

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
