# Peer review of "Characterization of Organosolv Lignins and Their Application in the Preparation of Aerogels"

_materials, 2022, doi:10.3390/ma15082861_

Round 1

Reviewer 1 Report

I recommend publishing the paper after some changes:

  1. Please reorganize the introduction part as it is hard to follow what the main aim of the paper is. Some information has been just written without following any storyline.
  2. The last paragraph of the introduction should strongly focus on the purpose of the work which is vague for the readers now.
  3. The difference between the present work and previous articles from other groups and authors should be mentioned in the introduction part. But, the lack of such comparison can be completely sensed.
  4. The definition of aerogels should be described at least in one sentence. The drying processes have been mentioned, while the concept of aerogels and their fabrication process was totally ignored.
  5. Did you utilize a well-known process for the extraction of lignin or a new one? What are the other appliable processes? What are the advantages of different processes? Why did you employ this process?

Some information has been written in the introduction part, but it is not enough to emphasize the applied process for the present work.

  1. It is better to mention what EOL and DOL are in the caption of Table 1.
  2. Figure 2 must be put after the related explanation. Please be careful about the axis title (complete title should be placed).
  3. The quality of Figures 3 and 4 is too low.
  4. How did the authors obtain bulk density for the aerogels? Please explain it.
  5. Please calculate porosity for each sample from skeletal and bulk densities.
  6. Figure 6, be careful about the axis title.
  7. Name figures and tables in the script before placing them in the manuscript.
  8. Indeed, the gel time of aerogels for the 85% replacement of 5-MR with lignin is high, but bulk density and dry shrinkage are almost lower for most of these aerogels (85% replacement). So, why did the authors choose aerogels with 75% replacement for further examinations?

Also, the gels time is not too high for some aerogels with 85% replacement, while their bulk densities are much lower (AG-EOL Pine and AG-DOL Pine).   

  1. Please improve the quality of figure 8. These figures show that the aerogels are so dense, explain it.

Reporting the porosity for them might be helpful as porosity is one of the parameters which can determine the difference between aerogels and foams. 

Reviewer 2 Report

This is a very nice and comprehensive work to reveal possibilities for the fabrication of aerogels based on using of organosolv lignins.

Different analytical approaches (IR, SEM) were used to accomplish exhaustive characterisation of the biomaterial.

Using any bioacceptable and biodegradable material for fabrication products that can replace hardly-disposable products does deserve an attention.

After examination, I have only two remarks to add:

1. In 8th paragraph of Intro, I miss to add a general statement of using biopolymers as green materials for gel fabrication.

“Biopolymer-based materials are suitable for fabrication of biopolymer-based gels [https://doi.org/10.1016/j.talanta.2020.121892]. Phenolic resins are well-known polymeric materials where by manipulating the phenolic (or substituted phenols) to aldehyde monomer ratio and amount of solvent, pH, catalyst type, reaction temperature, reaction time, give rise to a variety of resin structures with a wide range of possible properties.“

2. In Conclusion, please add some information about authors´ future aims in this research field.

Reviewer 3 Report

Luke and coworkers reported detailed characterizations of organosolv lignins from different types and their applications to the preparation of lignin-5-methylresorcinol-formaldehyde aerogels. Considering the great interest in novel materials and value-added chemicals from biomass such as lignin, this work is quite interesting since it provides with thorough structural characterizations such as the type of OH by 31P NMR, and aromatic ring and oxygenated aliphatic structures by HSQC NMR. They also prepared aerogels from the organosolv lignins and compared each other. This work seems worthy of publication in Materials after addressing the following issues.

  1. Introduction section seems informative and well described. However, compared to previous studies, it is not clearly described how original and how different this work is since many studies regarding lignin-based aerogels are already reported. The authors should specify those points in the Introduction.
  2. It would be helpful for readers if the chemical structures of three primary phenylpropanoid monomer units in lignin (i.e., S, G, and H units) are drawn.
  3. Thorough parametrization of organosolv lignins were conducted for different types of lignocellulosic biomass, but those parameters are not correlated with the properties of resulting aerogels. It seems very important to find out how differences in internal structures of organosolv lignins affect the properties of aerosols. Thus, the discursions for this part should be supplemented.
  4. The quality of figures can be improved. Some numbers/labels in the figures are too small to read. In addition, the authors may consider leaving only representative figures in the manuscript and moving the rest to Supplementary Information.
  5. The authors should carefully check typos and errors such as the headings of “Mw” and “Mn” in Table 1.

Round 2

Reviewer 1 Report

The authors have accurately addressed all issues. From my vantage point, the current version is ready for publication.